# Functional Yogurt Fortified with Honey Produced by Feeding Bees Natural Plant Extracts for Controlling Human Blood Sugar Level

**DOI:** 10.3390/plants11111391

**Published:** 2022-05-24

**Authors:** József Prokisch, Hassan El-Ramady, Lajos Daróczi, Éva Nagy, Khandsuren Badgar, Attila Kiss, Ayaz Mukarram Shaikh, Ibolya Gilányi, Csaba Oláh

**Affiliations:** 1Institute of Animal Science, Biotechnology and Nature Conservation, Faculty of Agricultural and Food Sciences and Environmental Management, University of Debrecen, Böszörményi út 138, 4032 Debrecen, Hungary; hassan.elramady@agr.kfs.edu.eg (H.E.-R.); b_khandsuren@muls.edu.mn (K.B.); 2Soil and Water Department, Faculty of Agriculture, Kafrelsheikh University, Kafr El-Sheikh 33516, Egypt; 3Y-Food Ltd., Dózsa György út 28/A, 4100 Berettyóújfalu, Hungary; daroczi@yfood.hu (L.D.); nagyevacska@gmail.com (É.N.); 4Knowledge Utilization Center of Agri-Food Industry, University of Debrecen, Böszörményi út 138, 4032 Debrecen, Hungary; attkiss@agr.unideb.hu; 5Institute of Food Science, University of Debrecen, Böszörményi út 138, 4032 Debrecen, Hungary; ayaz.shaikh@agr.unideb.hu; 6Department of Laboratory, Borsod County Teaching Hospital, 3526 Miskolc, Hungary; gilanyi.lab@bazmkorhaz.hu; 7Department of Neurosurgery, Borsod County Teaching Hospital, 3526 Miskolc, Hungary; olahcs@gmail.com

**Keywords:** active ingredients, acacia honey, chlorella alga honey, sea buckthorn honey

## Abstract

The human blood sugar level is important and should be controlled to avoid any damage to nerves and blood vessels which could lead to heart disease and many other problems. Several market-available treatments for diabetes could be used, such as insulin therapy, synthetic drugs, herbal drugs, and transdermal patches, to help control blood sugar. In a double-blind human study, four kinds of honey from bees fed on acacia, sea buckthorn, chlorella alga, and green walnut extracts were used in fortifying yogurt for controlling human blood sugar. The impact of a previously fortified honey was investigated on blood levels and other parameters of healthy individuals in a human study with 60 participants. The participants received 150 mL of yogurt mixed with 30 g of honey every morning for 21 days. Before and after the study period, the basic blood parameters were tested, and the participants filled out standardized self-report questionnaires. Acacia honey was the traditional honey used as a control; the special honey products were produced by the patented technology. The consumption of green walnut honey had a significant effect on the morning blood sugar level, which decreased for every participant in the group (15 people). The average blood sugar level at the beginning in the walnut group was 4.81 mmol L^−1^, whereas the value after 21 days was 3.73 mmol L^−1^. The total decrease level of the individuals was about 22.45% (1.08 mmol L^−1^). Concerning the sea buckthorn and chlorella alga-based honey product groups, there was no significant change in the blood sugar level, which were recorded at 4.91 and 5.28 mmol L^−1^ before treatment and 5.28 and 5.07 mmol L^−1^ after, respectively. In the case of the acacia honey group, there was a slight significant decrease as well, it was 4.77 mmol L^−1^ at the beginning and 4.27 mmol L^−1^ at the end with a total decrease rate of 10.48%. It could thus be concluded that the active ingredients of green walnut can significantly decrease the blood sugar level in humans. This study, as a first report, is not only a new innovative process to add herbs or healthy active ingredients to honey but also shows how these beneficial ingredients aid the honey in controlling the human blood sugar level.

## 1. Introduction

Diabetes is a common and serious metabolic disorder threatening public health worldwide as one of the top 10-deadly diseases in adults, causing 4 million deaths globally every year [1]. In March 2022, it was estimated that the global diabetes prevalence was 537 million people, causing at least 966 billion USD in health expenditure [2]. Additionally, it is expected that diabetes prevalence will rise from 578 to 700 million by 2030 and 2045, respectively [3]. A high blood sugar level is the “main known symptom” of diabetes, in addition to other symptoms such as increased urination, blurred vision, unexplained fatigue, increased thirst and hunger, and unexpected weight loss [4,5]. Controlling the blood sugar level in humans is important to avoid any damage to nerves and blood vessels due to the rise in carbohydrate contents (hyperglycemia) which leads to heart disease and many other problems [3,6]. Several approaches have been attempted to manage diabetes including medication and natural products such as honey or medicinal herbs [7].

About 300 different types of honey are well known (e.g., acacia, citrus, clover, honeydew, Majra, Manuka, mountain, Sidr, and Tualang honey) which are differentiated by their sources and their seasonal and geographical origin as well as their processing, harvesting, and storage conditions [8,9]. Honey is called ‘medical-grade honey’ because of its antibacterial activity, use in several medical treatments, such as diabetes, and promotion of human health as a functional food [10]. The use of honey in the treatment of diabetes mellitus was confirmed by several studies (e.g., [8,9,11,12,13,14,15,16]). For the potential role of honey in managing glycemia (human blood sugar), honey as a sweetener could be added to food, such as yogurt, which could be consumed daily for chronic diseases [17,18]. Honey has been fortified with many kinds of foods to improve their nutritional and medicinal attributes such as dairy products [19,20,21,22,23], various other kinds of foods [24], and licorice [25].

Functional foods are foods that are used for specific therapeutic purposes or human health because of their high content of bioactive compounds such as alkaloids, carotenoids, flavonoids, phenolic acids and compounds, stilbenes, and lignans, tannins, terpenes, and terpenoids [26,27]. Several nutrients or bioactive ingredients have been applied to different kinds of foods through the fortification process, such as producing functional foods with honey and/or yogurt, microalgae or chlorella alga [28,29,30], sea buckthorn [31,32,33], and walnut [34,35]. These functional foods have the potential for health-promoting attributes [36]. Dairy products (i.e., yogurt, milk, and cheese) are well-known for their health-enhancing attributes, especially yogurt, which can promote human antimicrobial activity, stimulate the immune system, and ameliorate the digestibility of proteins and lipids [23,24]. Therefore, many studies have focused on yogurt fortified with honey due to their benefits for human health, for example, increasing the antioxidant activity [23] and treating some health problems such as vulvovaginal candidiasis [37]. Some studies reported on the feeding of honey bees using the syrup of natural plant extracts such as the extract of algae *Spirulina platensis* [38], but to the best of our knowledge, there are no reports about yogurt fortified with honey from bees fed on the natural plant extracts studied.

Therefore, natural plant extracts (i.e., acacia, green walnut, sea buckthorn, and chlorella alga) were added to yogurt fortified with honey, which means honeybees were fed on these plant extracts, to manage and control the human blood sugar level. This study is an attempt, as a first report, to answer the main question: which plant extract reports the best results in decreasing human blood sugar levels?

## 2. Results

The current study was carried out to investigate which plant extracts from the tested candidates can decrease the morning blood sugar level in a clinical trial. The collected blood samples during the clinical experiment were stored in a refrigerator at 4 °C until measuring the basic chemical composition of blood before and after consumption (Appendix A). The measured parameters included glucose, triglycerides, cholesterol, creatinine, immunoglobulin, uric acid, ferritin, albumin, transferrin, and blood serum such as Ca, Mg, and Fe (Appendix A). Blood sugar level was measured in all participants in each group and for each honey group treatment as well. Concerning the first group (acacia honey as a control), the blood sugar levels decreased from the participants consuming the acacia honey fortified yogurt but not in a sharp relationship (Figure 1). For the second and third groups, the relationship between blood sugar levels and consumed yogurt fortified with honey was not clear and not significant, whereas, for the fourth group (walnut), all values of participant blood sugar after consuming the yogurt fortified with honey were lower than values after getting treatments (Figure 1).

The basic analyses of the honey for the four groups are listed in Table 1 From this table, it is shown that the chemical content of the honey for each group, in general, is similar and there are no big differences among the groups including the sugar content (fructose + glucose was higher in the case of chlorella alga) and different measured nutrients in the honey. Eleven nutrients were also measured in the honey before applied treatments including the contents of B, I, Cu, Ca, Mg, K, and Zn.

Based on the statistical analysis for the studied groups of honey, a significant decrease in morning sugar blood level was observed in the walnut group (4) (Figure 2), whereas groups 2 (Green alga) and 3 (Sea buckthorn) did not record any significant difference, although the four groups had the same lower level of morning blood sugar (less than 6). Concerning group 4, the values of morning blood sugar in the volunteers before the applied treatments were 4.8 and were 3.7 mmol L^−1^ after, as presented in Figure 2. The data were presented in ascending order in Figure 3, where the presented values were calculated by subtraction of the morning blood sugar value measured on the 21st day of the trial from the day 0 of the trial. Almost all of the participants had lower blood sugar levels. The average decreasing rate was 22.45% for the group of walnut, namely, 1.07 mmol L^−1^, and the maximum was 2.5 mmol L^−1^ (6.3–3.8 mmol L^−1^).

## 3. Discussion

This study has a patented process, which did not represent feeding the bee colonies with sugar/sugar syrup during the barren period (when natural bee pastures do not provide enough food), to feed bees with special herbs or the immune-boosting natural active ingredients of plants or minerals. These natural active ingredients have a beneficial physiological effect on bees and are excreted in honey, which has a beneficial effect on people consuming honey, and can cause immune-boosting and many beneficial physiological effects. This study also confirmed that honey could be consumed with yogurt as a *functional yogurt fortified with honey*. These functional preventive honey products could be developed as functional food produced by bees and the number of active components in them could significantly exceed the useful ingredients in natural kinds of honey. The “oriented production” of honey could be controlled to be enriched with special active ingredients by incorporating these ingredients into the honey by feeding the bees these ingredients, such as feeding honeybees with some herbs [39], plant extracts of mint, cinnamon, and chamomile [40], or feeding with *Spirulina platensis* extract to increase the honey’s content of antioxidants and phenolics [38].

Using this strategy, three natural plant extracts were selected in this study to produce honey rich in active ingredients, which are already involved in these extracts, and to preserve the health of bees and humans as well. This makes the honey product economically competitive compared to other traditional methods in which the active ingredients are processed from natural sources. Thus, based on analytical studies, it has been established that the physiologically beneficial groups of active substances that can be detected in the herb, on average about 80%, are also transferred to the sugar syrup and, after conversion by the bees, to the honey-like bee product. Thus, there is an urgent need for controlling the feeding process of bees as an effective method to enrich honey with desirable bioactive plant components, which can be achieved using natural herbs. This method is considered a creative and innovative way of producing bee products rich in bioactive compounds as reported by recent studies (e.g., [41,42,43,44]). These bioactive compounds mainly depend on the plant species, such as the coumarin in the case of using the flowers of the *Melilotus officinalis* herb [42], or patented herbal extract [44]. This current study is a new and patented technology, crucial as a new solution for supporting beekeepers and their feeding of bees with a preservative material as compared to the known beekeeping technologies. 

The production of yogurt rich in bioactive ingredients for the human diet is an important approach in the field of food production as reported by many studies using *Nyctanthes arbor-tristis* L. flower extract [45], fennel seed extract [46], and nutmeg, white and black pepper [47]. These studies depend on the fortification approach to produce yogurt rich in bioactive compounds from natural herbs in order to improve yogurt’s nutritional and health benefits. In the current study, the yogurt was fortified with honey produced by underfeeding bees with the extract of three natural plants (i.e., green algae, sea buckthorn, and green walnut compared to acacia, as a control) for diabetes treatment. The yogurt fortified with honey produced for each group could be discussed separately for each natural plant extract. Concerning acacia honey, it is very common, and of high quality in Hungary in addition to oilseed, rape, sunflower, and forest. Hungary is considered one of the highest honey producers in the European Union [48,49]. In the current study, acacia honey had the highest water content among the studied honey from other plant extracts (19.2%) but had the lowest content of hydroxy-methyl-furfural (HMF) content 1.73 mg kg^−1^ and fructose + glucose content of 64.7%, as well as lowest values of the following nutrients (in mg kg^−1^): calcium (62.8), potassium (142), sodium (14), phosphorus (68.5), sulfur (19), and zinc (0.86), compared to other treatments. 

In regards to the relationship between honey and diabetes, many reports on honey and its anti-diabetic activities have been confirmed by researchers such as Sharma et al. [8]. They reported the mechanism of honey (proposed antibiotic effects), which includes the regulation of the activity of pancreatic β-cells and related hormones, kidney, liver, eye, intestine, nerve, gut, muscle, and vasculature through antioxidant, antimicrobial, antihypertensive, anti-inflammatory, immunomodulatory, wound-healing, hypolipidemic, hypoglycemic, and nutritional effects. Honey also has a low glycemic index due to its main sugars including fructose and glucose in addition to its high content of amino acids, minerals, enzymes, phenolic compounds, and vitamins, which may manage diabetes depending on the honey source, its composition, and administered dose [8,50].

The great potential of chlorella alga as microalgae have been demonstrated as a sustainable food supply to meet the population’s needs because of their considerable phyto additives and/or bio-active phytonutrients, lipid, carotenoid, and protein content [28,29,51]. Many studies have reported on using microalgae for therapeutic potential and their applications (e.g., [52]), including diabetes [53]. Chlorella alga, as a unicellular microalga, is a popular food in different countries, particularly in East Asia (e.g., China, Korea, Indonesia, Japan, and Taiwan), due to its nutritional value as a relatively complete food [54]. In the current study, the role of green algae in reducing the blood sugar level was not significant, as found in Figure 4.

Sea buckthorn belongs to the family Elaeagnaceae, and its berries are associated with reducing the risk of many human diseases such as those that are cardiovascular [55]. This plant also has a distinguished potential for reducing blood glucose as an antidiabetic agent through glycemic control [32,56]. In human trials, applying 40 g of dried sea buckthorn to 200 g yogurt and 50 g glucose has been suggested to suppress peak insulin response and stabilize postprandial hyperglycemia [32]. This plant is also rich in many bioactive compounds including flavonoids and fat-soluble vitamins such as A, E, and K [32]. Although, the role of sea buckthorn in reducing the blood sugar level was confirmed by many studies (e.g., [32,55]), the current study showed no significant difference before and after applying yogurt fortified with honey from bees fed on sea buckthorn extract (Figure 4). 

Green walnuts were selected as young or immature walnuts because they are not yet allergic, have a high content of vitamin C (more than oranges), iron, and tannins, and are extremely rich in phenolic compounds. Walnut fruits can provide a wide range of nutrients including vitamins (E, B3, B9, B6, etc.), minerals (Ca, Mg, K, etc.), and many other bioactive compounds, such as antioxidants, phytosterols, and phenolic compounds [57]. This plant is also cultivated for products in cosmetic, pharmaceutical, and agricultural industries, which includes the green seeds, husks, kernels, shells, bark, and leaves, as it is a very rich source of ascorbic acid, phenolics, and tocopherols [58]. Walnut has a hypoglycemic herbal effect with lowering the blood glucose for diabetes mellitus, as confirmed by using walnut seeds [59] and leaves [58] due to the impact of phenolic compounds and fatty acid contents. 

In the current study, a sharp, significant effect of green walnut honey was observed in reducing human blood sugar levels (Figure 4). The average blood sugar level at the beginning in the walnut group was recorded at 6.3 mmol L^−1^ and decreased to 3.8 mmol L^−1^ after 21 days of the treatments. The decrease level of individuals was 2.5 mmol L^−1^ (by the rate of 39.7%), the highest decrease in blood sugar level compared to all other treatments (*Acacia*, *Sea buckthorn*, and *Chlorella* alga honey groups). The most distinguished parameter in green walnut honey is the HMF content, which was the lowest value (4.88 mg kg^−1^) among other treatments (second and third group). The HMF value is considered an important indicator of the quality of honey as reported by Shapla et al. [60].

## 4. Materials and Methods

### 4.1. Preparation of Plant Extracts for Honeybee Feeding

The syrup of three different plant extracts was added for feeding honeybees, which was then used in fortified yogurt to decrease the level of human blood sugar. Acacia (*Acacia penninervis* L.) plants, as a control and for traditional honeybee breeding without any criteria for feeding, were obtained from the primary producer Ferenc Veres in Berettyóújfalu in Nyírség, on the border of Mándok city (Hungary). The plant extracts included three different extracts collected from different places in Hungary as follows: (1) green algae or chlorella alga (*Chlorella* sp.) were obtained from Albitech Kft (Budapest, Hungary), (2) sea buckthorn (*Hippophae rhamnoides* L.) was harvested from the organic garden of Orbán Fruzsina (Debrecen), and (3) green walnut (*Juglans regia* L.) was harvested from the organic garden of beekeeper János Sáfián (Debrecen, Hungary). The basic information on the plant extracts used is presented in Table 1. The previously mentioned extracts were used for preparing feeding syrups, which were prepared by the following procedure, and then used to prepare honey products from each feeding syrup as the first step (Figure 4 and Figure 5). 

The traditional method for honey production starts with the bees visiting plant flowers to collect the nectar. Once the nectar has been extracted from the flowers, the bees digest the nectar using enzymes, converting it into simpler compounds, mainly fructose and glucose, and storing it in their abdomen. The general principle with the syrup making was that the raw plant material was mixed with five times more sugar. Consequently, water was added to the sugar-plant mixture. The final concentration of sugar was about 50% in the syrup; the exact amount was determined by the density of the syrup. The density and the pH have an important role in the syrup. According to many years of experience, beekeepers suggest a 1.23 g cm^−3^ density value. If it is higher, then the bee has difficulties in the uptake of the syrup, if it is lower, then the fermentation would be very intensive which changes the pH and the bees would likely avoid it. The pH can be set with partially fermented syrup if it is necessary. The inventors tested, produced, and commercialized more than 10 different plant extract-based honey. The three selected products were picked out according to consumer satisfaction and the experiences of the beekeepers. If these types of syrups were preferred by the bees, it would be possible to increase the production values. The final products have the highest sale in the market and they had the best consumer satisfaction.

For the administration of syrup, a special feeding technique was used. A sealed thin polyethylene (PE) bag was filled with the syrup and placed under the top of the beehive. The sucker of the bee can penetrate the PE bag and the bee can suck out the syrup. The bag is air-tight, sealed, and its mass can be measured easily. With this method, the infection or contamination of syrup can be avoided.

### 4.2. Preparation of Yogurt Products

Yogurt inoculated by heat-treated technology was put into a plastic bucket from the aging tank. After determining the amount of raw material, the measured honey and bee products were added. After homogenization, the mixed semi-finished product was poured into the ballast tank of the beaker. The product was then dispensed into a plastic cup in the circular dosing machine, sealed with welded aluminum foil, and collected in a collection tray containing 20 pieces. The sample packaged in this way was aged in a refrigerator for 24 h at 4–6 °C ((Figure 6) as an example for preparing green walnut honey). The freezing chain of the polystyrene refrigerator was extended to the place where human clinical trials are conducted at the Borsod-Abaúj-Zemplén County Central Hospital and the University Teaching Hospital (Miskolc city, Hungary). The chemical composition of the honey used before applying treatments from the different groups was measured as listed in Table 2. Concerning the methodology of the parameters in Table 2, the basic parameters including water content, fructose + glucose, fructose, glucose, free acid content, HMF content, and diastase activity were determined according to the standard methods of the International Honey Commission [61], whereas the determination of nutrient concentrations (i.e., boron, calcium, copper, iron, potassium, magnesium, sodium, phosphorus, sulfur, and zinc) was measured by ICP-OES (Inductively Coupled Plasma Optical Emission Spectrometer) (Thermo Scientific iCAP 6300, Cambridge, UK) according to Czipa et al. [62].

### 4.3. The Protocol of the Human Study

A total of 60 participants aged 24–55 years participated in the clinical study, of which, 30% were male and 70% were female, and were divided into four groups with each group having 15 separated (not repeated) participants. Participants were selected based on the following criteria and were repeatedly followed by a flow chart (Table 3). Selection criteria were carried out based on the following items: a signature on a statement of consent and healthy male or female volunteers over the age of 18. Exclusion criteria included: (1) history of stroke, severe cerebrovascular accident, (2) history of acute myocardial infarction, (3) within six months of surgery, (4) pregnancy or breast-feeding, (5) cancerous disease, (6) severe acute immunological or pulmonological disease, (7) history of multiple drug sensitivity, (8) known milk protein allergy, lactose intolerance, (9) celiac disease, gluten-sensitive enteropathy, Duhring disease, (10) Crohn’s disease, ulcerative colitis, (11) flower powder allergy, (12) thyroid disease, and (13) diabetes, high blood sugar level. The main steps that were carried out during the current study are presented in Figure 7.

### 4.4. Sampling

Blood samples were taken by qualified personnel with maximum adherence to and ensuring sterility rules on the 1st day and the 21st day of the study; biochemical and hematological parameters of blood were measured using the ADVIA Chemistry XPT System (Siemens Medical Solutions Inc., Malvern, PA, USA). Blood samples were analyzed and the following chemical tests were performed: IgGAM, and IgE, Sysmex 1000 (Sysmex, Kobe City, Japan) for blood count, Adams A1c HA-8180V (Arkray, Kyoto City, Japan) for HgbA1c, and Hydrasys (Sebia, Lisse, France) agarose gel electrophoresis. The results of this study did not include any repeating during the meaning processes.

### 4.5. Ethical Permission

The clinical trial is a double-blind, randomized, controlled follow-up pilot study. The test was started and performed with RKEB Ethics Committee’s permission. The number of ethical permission is IG-50-102/2019.

### 4.6. Statistical Analyses

Laboratory results were evaluated by an analysis of variance *t*-test and one-way ANOVA using the SPSS statistical program, SPSS V22.0, New York, the United States [63].

## 5. Conclusions

Honey is a very important functional food for human health because of its high content of anti-microbial activity and for several medical treatments; in particular, diabetes mellitus. Honey is a very common sweetener that could be used to manage human blood sugar (glycemia) or/and for chronic diseases when added to many foods such as yogurt. The production of functional yogurt fortified with honey enriched with active compounds from four natural plant extracts was the first target of this study as a new technology. This fortified yogurt was consumed by 60 participants who suffer from diabetes for 21 days. The morning blood sugar level was measured daily during the 21 days for the 60 participants with diabetes. Green walnut plant extract recorded the best results in decreasing human blood sugar levels compared to other extracts. The results confirm the crucial role of green walnut in treating diabetes as a preliminary human study and are the first report including new technology. This study has opened many new windows toward a new approach to diet fortification using yogurt and honey. The new approach of the “oriented production of honey” to be rich in certain bioactive ingredients from natural plant extracts is a promising new technology that needs more study. What is the further research after producing this yogurt fortified honey fed on natural plant extracts? Which bioactive compounds will be dominant using different types of honey? To what extent will these bioactive ingredients be useful under higher concentrations of applied plant extracts? Can this new technology be used for the biofortification of nano nutrients such as nano Se?

## Figures and Tables

**Figure 1 plants-11-01391-f001:**
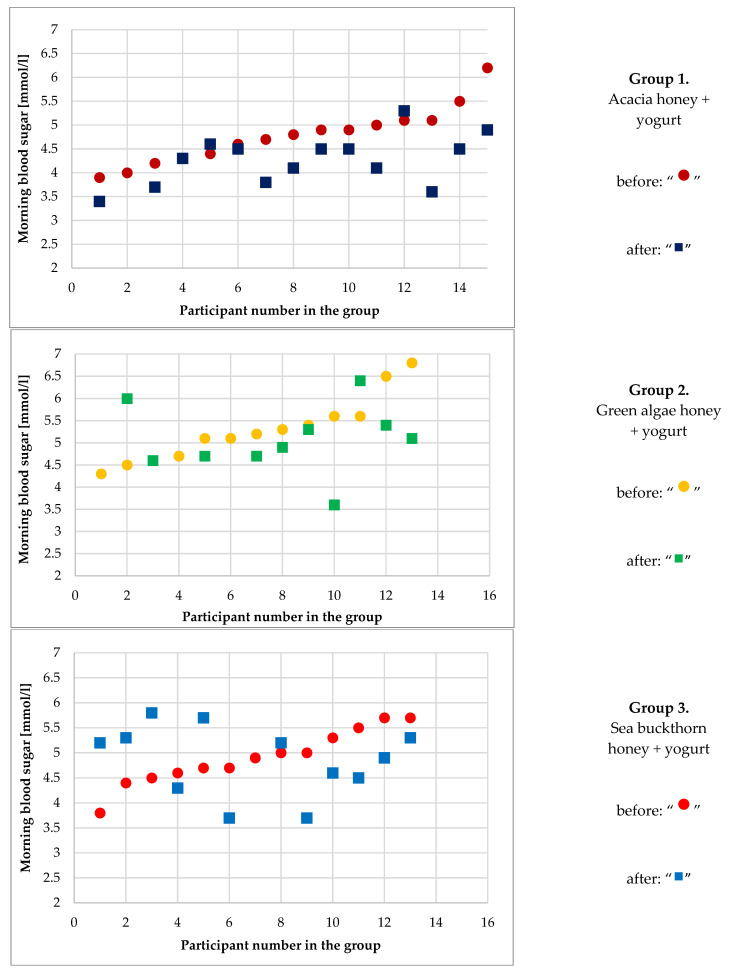
The blood sugar level of each participant in the four treatment groups of the clinical trial before (day 0) and after (day 21) of the trial. The participants received 30 g of four different honey products in 150 g of yogurt over 21 days. Some volunteers were excluded from the study from the first, second, and third groups because one had a high initial blood sugar level and the others gave up participation in the trial. In the fourth group, all of the participants took part from the beginning until the end.

**Figure 2 plants-11-01391-f002:**
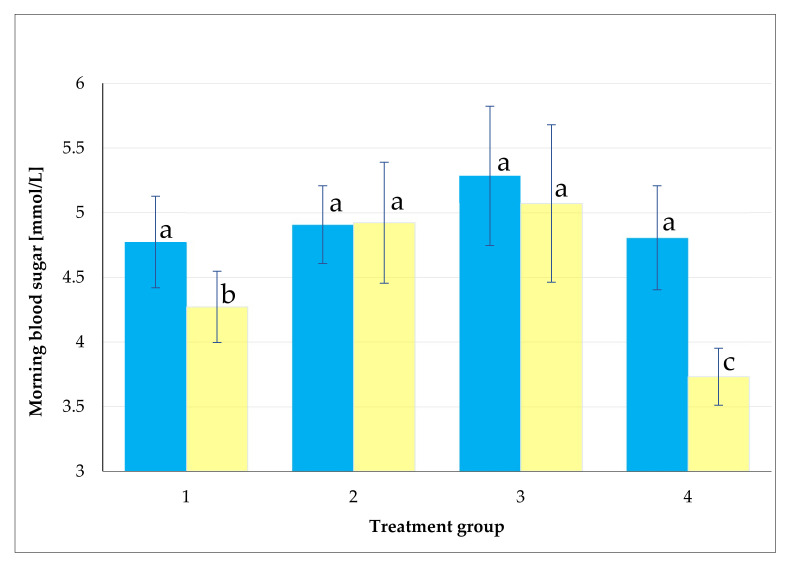
Mean values of morning blood sugar of the participants before (blue column) and after (yellow column) the human clinical trial. The treatment groups included group (1) acacia honey + yogurt; group (2) green algae honey + yogurt; group (3) sea buckthorn honey + yogurt; and group (4) walnut honey + yogurt. The participants received 30 g of honey product in 150 g of yogurt for 21 days. The blood samples were taken before and after the 21-day trial. The same letters mean the values were not significant at 5%.

**Figure 3 plants-11-01391-f003:**
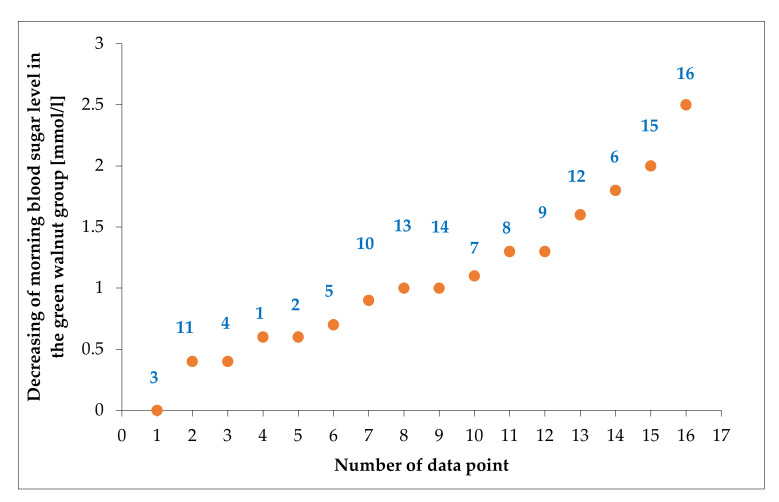
The decrease of morning blood sugar values in the blood of volunteers in treatment group 4. The data were presented in ascending order and calculated by subtraction of the morning blood sugar value measured on the 21st day of the trial from day 0 of the trial.

**Figure 4 plants-11-01391-f004:**
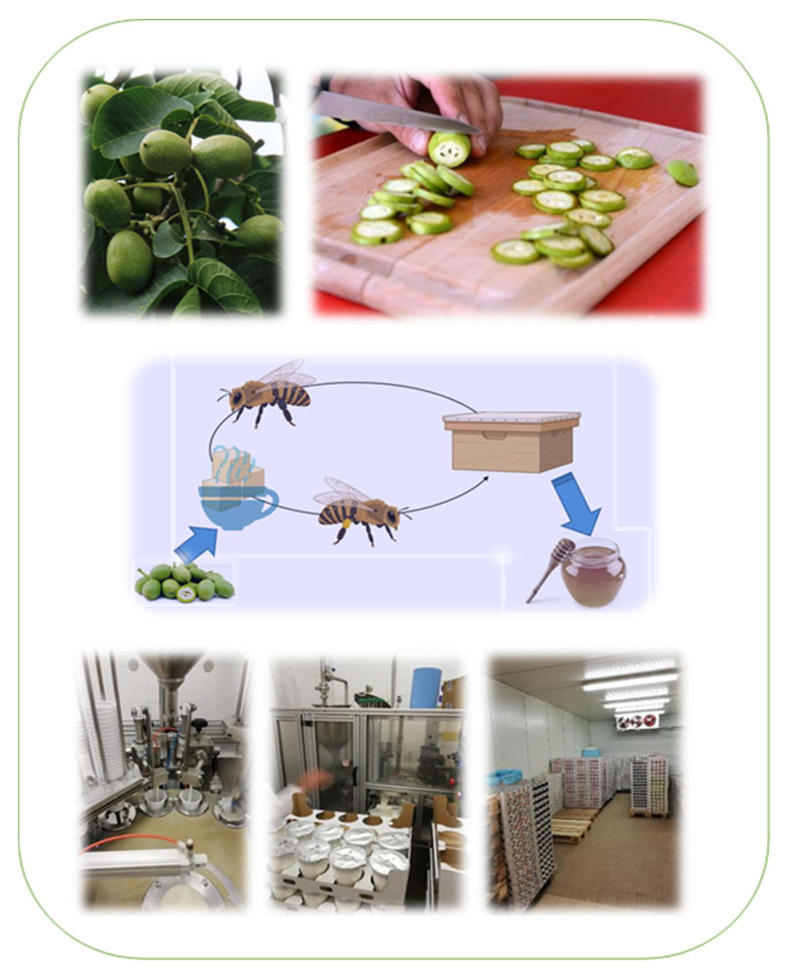
Different steps in producing the honey yogurt for the clinical trial. First, the green walnuts were collected (upper photos), selecting the desired part from the green fruits to make the syrup to feed the bees (middle photos). After producing the honey from green walnuts, it was mixed with yogurt in the packaging in the factory (lower photos).

**Figure 5 plants-11-01391-f005:**
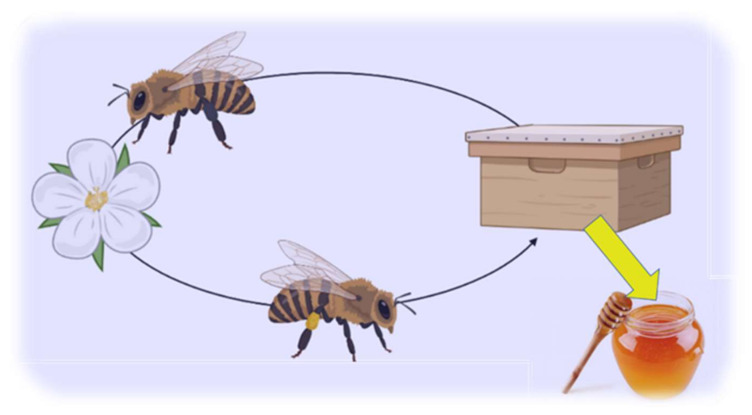
The traditional method for honey production starts with the bees visiting plant flowers to collect the nectar. Once the nectar has been extracted from the flowers, the bees use enzymes to digest the nectar into simpler compounds, mainly fructose and glucose, and store it in their abdomen.

**Figure 6 plants-11-01391-f006:**
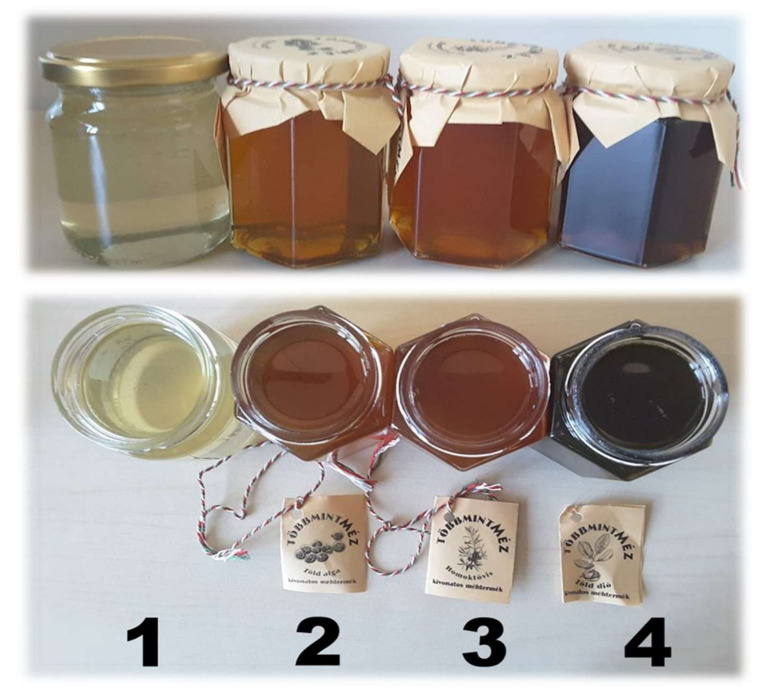
The tested honey in the clinical products belonging to each tested group includes (1) acacia, (2) chlorella alga, (3) sea buckthorn, and (4) green walnut.

**Figure 7 plants-11-01391-f007:**
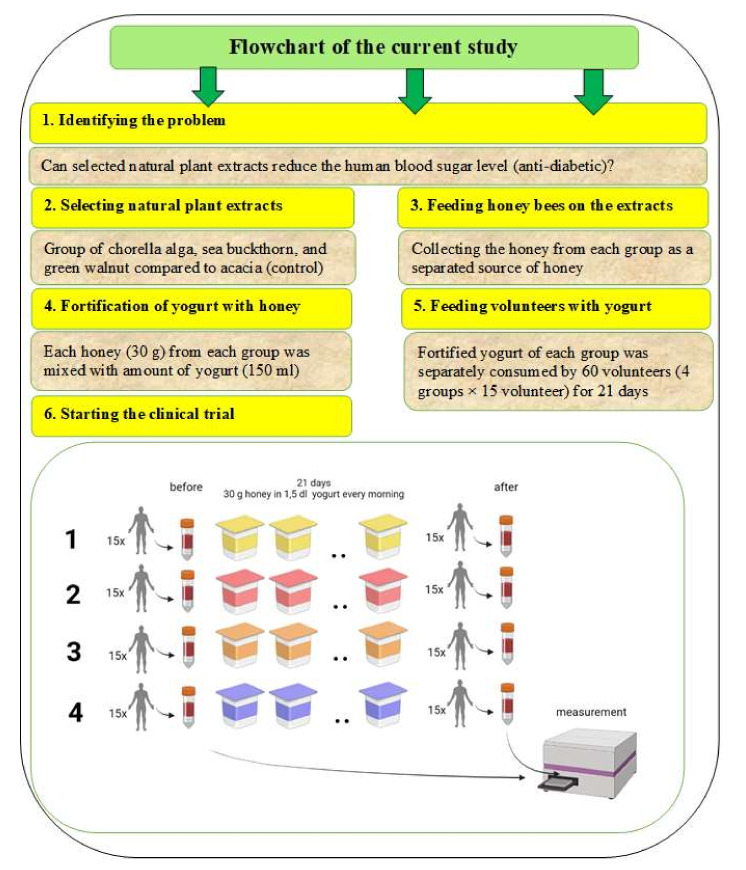
The main steps already carried out during this study include the clinical trial. The tested honey products were consumed in yogurt. The treatment: (1) acacia honey (as a control), (2) chlorella alga honey, (3) sea buckthorn honey, and (4) green walnut honey. In total, 30 g of honey product was added to 150 mL of yogurt. The volunteers consumed only one yogurt-honey box every day for 21 days and blood samples were taken before and after the yogurt consumption during the clinical trial. **Very important note:** Some volunteers were excluded from the study as follows: for the first, second, and third groups, one, three, and one volunteer, respectively. One had a high initial blood sugar level, and the others gave up participation in the trial, whereas, in the fourth group, all of the participants took part from the beginning until the end.

**Table 1 plants-11-01391-t001:** Composition of the tested honey of selected plant extracts (i.e., chlorella alga honey, sea buckthorn honey, and green walnut honey) compared to the acacia honey used before treatments.

Measured Parameter/Nutrient	Unit	Acacia Honey (Control)	Chlorella Alga honey	Sea Buckthorn Honey	Green Walnut Honey
Water content	% (m/m)	19.2 ± 0.1 ^a^	18.5 ± 0.1 ^b^	18.6 ± 0.1 ^b^	19.0 ± 0.1 ^c^
Fructose + glucose	% (m/m)	64.7 ± 0.20 ^a^	69.1 ± 0.20 ^b^	65.9 ± 0.6 ^c^	65.0 ± 0.6 ^ac^
Fructose	% (m/m)	34.9 ± 0.4 ^a^	35.4 ± 0.5 ^a^	41.0 ± 0.4 ^b^	36.2 ± 0.2 ^c^
Glucose	% (m/m)	29.8 ± 0.3 ^a^	33.7 ± 0.4 ^b^	24.9 ± 0.3 ^c^	28.8 ± 0.3 ^d^
Free acid content	mmol L^−1^	23.5 ± 0.4 ^a^	37.5 ± 0.6 ^b^	75.6 ± 0.4 ^c^	45.3 ± 0.2 ^d^
HMF content	mg kg^−1^	1.73 ± 0.20 ^a^	55.6 ± 2.0 ^b^	42.3 ± 2.0 ^c^	4.88 ± 0.3 ^d^
Diastase activity	Goethe number	6.07 ± 0.10 ^a^	4.17 ± 0.12 ^b^	<4.0 ^c^	8.68 ± 0.1 ^d^
Boron (B)	mg kg^−1^	3.12 ± 0.20 ^a^	2.14 ± 0.23 ^b^	0.71 ± 0.12 ^c^	2.09 ± 0.19 ^d^
Calcium (Ca)	mg kg^−1^	62.8 ± 2.9 ^a^	125 ± 13 ^b^	131 ± 13 ^b^	120 ± 26 ^b^
Copper (Cu)	mg kg^−1^	0.45 ± 0.11 ^ab^	0.46 ± 0.03 ^a^	0.36 ± 0.02 ^b^	0.56 ± 0.09 ^a^
Iron (Fe)	mg kg^−1^	<0.10 ^a^	0.68 ± 0.11 ^b^	1.60 ± 0.15 ^c^	1.28 ± 0.04 ^d^
Iodine (I)	mg kg^−1^	<0.10 ^a^	<0.10 ^a^	310 ± 10 ^b^	<0.10 ^a^
Potassium (K)	mg kg^−1^	142 ± 9.0 ^a^	415 ± 8.0 ^b^	428 ± 41 ^b^	321 ± 8.0 ^c^
Magnesium (Mg)	mg kg^−1^	2.75 ± 0.05 ^a^	15.7 ± 0.1 ^b^	27.0 ± 1.7 ^c^	15.5 ± 0.7 ^b^
Sodium (Na)	mg kg^−1^	14.0 ± 1.6 ^a^	55.4 ± 2.1 ^b^	49.6 ± 3.1 ^c^	51.9 ± 4.6 ^bc^
Phosphorus (P)	mg kg^−1^	68.5 ± 4.9 ^a^	113 ± 3 ^b^	79.9 ± 6.4 ^a^	89.6 ± 0.7 ^c^
Sulfur (S)	mg kg^−1^	19.0 ± 1.5 ^a^	38.2 ± 0.4 ^b^	36.8 ± 1.9 ^b^	38.7 ± 0.5 ^b^
Zinc (Zn)	mg kg^−1^	0.86 ± 0.04 ^a^	1.53 ± 0.12 ^b^	3.55 ± 0.25 ^c^	2.39 ± 0.25 ^d^

Abbreviation: Hydroxy-methyl-furfural (HMF). The ^a–d^ letters show the Duncan test result. The same letters mean there is no significant difference between the numbers in the row.

**Table 2 plants-11-01391-t002:** Basic information about the preparation of syrup for honey production in each group.

**Item in Detail**	**Group of Green Algae**	**Group of Sea Buckthorn**	**Group of Green Walnut**
Syrup preparing	Water and sugar	Water and sugar	Water and sugar
- Total volume †	50 L	24 L	400 L
- Extract, kg or L	15 L	8 kg	40 kg
Feeding rate	5 L per day	2 L per day	5 L per day
Feeding period	8 days	12 days	20 days
Syrup density	1.20 kg L^−1^	1.25 kg L^−1^	1.20 kg L^−1^
Syrup pH (start)	5.8	4.3	4.6
Syrup pH (end)	5.2	3.9	3.9

Notes: † feeding rate per bee colony.

**Table 3 plants-11-01391-t003:** The main criteria used in the current study for selecting the participants.

	Baseline	3 Weeks	3 Months
Signing a statement of consent	✓		
Recording of anamnesis, somatic status	✓	✓	
Blood pressure, heart rate, weight control	✓	✓	
SF 36 questionnaire	✓	✓	
EuroQol EQ-5D Quality of Life Questionnaire	✓	✓	
EORTC QLQ-C30 Questionnaire	✓	✓	
Culinary questionnaire		✓	
Detection of sleep disorders	✓	✓	
Detection of viral infections	✓	✓	✓

Notes: The participants were given functional yogurt once daily for 21 days from the following groups of honey products: (1) green walnut, (2) green algae, (3) sea buckthorn, and (4) acacia, with 20 g of each previous extract in 150 mL natural yogurt. During the study, the participants were continuing their former normal lifestyle. We did not ask them for any dietary or lifestyle changes. General parameters and records of their medical status were examined. At the end of the study, the participants completed a simple culinary questionnaire about the taste of the product as well.

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
