# Peer review of "Functional Yogurt Fortified with Honey Produced by Feeding Bees Natural Plant Extracts for Controlling Human Blood Sugar Level"

_plants, 2022, doi:10.3390/plants11111391_

Round 1
Reviewer 1 Report
The manuscript "Feeding Bees on Natural Plant Extracts to Produce Functional Yogurt Fortified with Honey for Controlling Human Blood Sugar Level" focus on the use of fortified honey to control human blood sugar level. The aim of the study has merits, however, the results obtained are not enough informative, the experimental design and data analysis are not well-structured. The manuscript lack of key information in material and method section (eg. syrup obtainment).
Introduction
Line 85-89 These questions were not addressed in this study and should be removed from the introduction. Maintain only the aims of the present study.
Discussion
The authors might use the discussion also to describe the study limits and pitfalls also in relation to other studies. Line 154: please consider to add some references to discuss these sentences.
Material and Methods
Line 218: On which basis were selected these 4 different plant extracts? How was obtained the plant extract to fed bees? Actually, the extracts considered are only three since acacia is the control honey
Table 1
I suggest to remove acacia since it is not a syrup but a control honey
What does this symbol mean ‡? I cannot find note in the Table.
Why were considered different feeding period?
did the final ph influence the quality/appetability of the syrup?
why the total volume of syrups is high variable? I suppose it is related to the number of the bees in a colony.
Figure 4
Please consider to add some details on the administration of syrup to colonies (bottle feeder....)
Figure 6
The figure is not clear enough to describe the process. It seems that green walnut is directly added in the yoghurt and not used to prepare the syrup to fed bees.
Table 2
HFM is formed in sugar containing food when heated of cooked. Why it is so high the level in 2 and 3 honey? Are these levels dangerous for consumption? The Codex Alimentarius Standard commission has set the maximum limit for HMF in honey at 40 mg/kg (with a higher limit of 80 mg/kg for honeys originating from tropical regions) to ensure that the product has not undergone extensive heating during processing and is safe for consumption. The abbreviation of HMF content is only HydroxyMethylFurfural, remove “The Role of….”
The manuscript "Feeding Bees on Natural Plant Extracts to Produce Functional Yogurt Fortified with Honey for Controlling Human Blood Sugar Level" focus on the use of fortified honey to control human blood sugar level. The aim of the study has merits, however, the results obtained are not enough informative, the experimental design and data analysis are not well-structured. The manuscript lack of key information in material and method section (eg. syrup obtainment).
Introduction
Line 85-89 These questions were not addressed in this study and should be removed from the introduction. Maintain only the aims of the present study.
Discussion
The authors might use the discussion also to describe the study limits and pitfalls also in relation to other studies also considering the questions that the authors added in the introduction. Line 154: please consider to add some references to discuss these sentences.
Material and Methods
Line 218: On which basis were selected these 4 different plant extracts? How was obtained the plant extract to fed bees? Actually, the extracts considered are only three since acacia is the control honey
Table 1
I suggest to remove acacia since it is not a syrup but a control honey
What does this symbol mean ‡? I cannot find note in the Table.
Why were considered different feeding period?
did the final ph influence the quality/appetability of the syrup?
why the total volume of syrups is high variable? I suppose it is related to the number of the bees in a colony.
Figure 4
Please consider to add some details on the administration of syrup to colonies (bottle feeder....)
Figure 6
The figure is not clear enough to describe the process. It seems that green walnut is directly added in the yoghurt and not used to prepare the syrup to fed bees.
Table 2
HFM is formed in sugar containing food when heated of cooked. Why it is so high the level in 2 and 3 honey? Are these levels dangerous for consumption? The Codex Alimentarius Standard commission has set the maximum limit for HMF in honey at 40 mg/kg (with a higher limit of 80 mg/kg for honeys originating from tropical regions) to ensure that the product has not undergone extensive heating during processing and is safe for consumption. The abbreviation of HMF content is only HydroxyMethylFurfural, remove “The Role of….”
Author Response
The authors would like to thanks for Reviewer comments that improve our manuscript quality.
Comments and Suggestions for Authors
The manuscript "Feeding Bees on Natural Plant Extracts to Produce Functional Yogurt Fortified with Honey for Controlling Human Blood Sugar Level" focus on the use of fortified honey to control human blood sugar level. The aim of the study has merits, however, the results obtained are not enough informative, the experimental design and data analysis are not well-structured. The manuscript lack of key information in material and method section (e.g., syrup obtainment).
Response: the MS has several improvements in almost all sections of the MS. Syrup properties and more details in Table 1 were added to the revised MS from line 252 to 272, thanks!
The general principle was at the syrup making that the raw plant material was mixed with 5 times more sugar. Consequently, water was added to the sugar-plant mixture. The final concentration of sugar was about 50% in the syrup the exact amount was determined by the density of syrup. The density and the pH have important role in the syrup. According to many years’ experiences of beekeepers suggest 1.23 g cm-3 density value. If it is higher, than the bee has difficulties to uptake the syrup, if it is lower than the fermentation will be very intensive what change the pH and they would like to avoid it. The pH can be set with partially fermented syrup if it is necessary.
Introduction
Line 85-89 These questions were not addressed in this study and should be removed from the introduction. Maintain only the aims of the present study.
Response: ok, done, thanks!
Discussion
The authors might use the discussion also to describe the study limits and pitfalls also in relation to other studies. Line 154: please consider to add some references to discuss these sentences.
Response: Added to the revised MS, thanks!
Material and Methods
Line 218: On which basis were selected these 4 different plant extracts?
The inventors tested, produced and commercialized more than 10 different plant extract-based honey. The 3 selected products were picked out according to the consumer satisfaction and the experiences of the beekeepers. This type of syrups was preferred by the bees, it was possible to increase the production values. The final products have the highest sale on the market, and they had the best consumer satisfaction.
The added part was inserted into the revised MS in line from 252 to 264.
How was obtained the plant extract to fed bees?
Response: all information about the plant extracts were mentioned in details in the MS in the section “2.1 Preparation of plant extracts for honey bees feeding.”
Actually, the extracts considered are only three since acacia is the control honey
Response: yes, they were corrected, thanks!
Table 1
I suggest to remove acacia since it is not a syrup but a control honey
Response: yes, corrected, thanks!
What does this symbol mean ‡? I cannot find note in the Table.
Response: yes, removed, thanks!
Why were considered different feeding period?
Response: thanks! Because it is related to the number of the bees in a colony!
The feeding period was determined by the weather, temperature and rain and available flowers. The syrup was applied during the flowerless period.
did the final pH influence the quality/appetability of the syrup?
Response: The pH has effect on the appetability of the syrup. The highest consumption of syrup was between 4,2-4,4 pH values. It was determined from the consumed amount of syrup.
why the total volume of syrups is high variable? I suppose it is related to the number of the bees in a colony.
Response: Yes, the amount of syrup was determined by the number of bee colonies. thanks!
Figure 4
Please consider to add some details on the administration of syrup to colonies (bottle feeder....)
Response: For the administration of syrup a special feeding technique was used. A sealed thin polyethylene bag was filled with the syrup and it was placed under the top of beehive. The sucker of bee can penetrate the PE bag and the bee can suck out the syrup. The bag is air tight, closed and its mass can be measured easily. With this method the infection or contamination of syrup can be avoided.
Figure 6
The figure is not clear enough to describe the process. It seems that green walnut is directly added in the yoghurt and not used to prepare the syrup to fed bees.
Response: yes, the figure is corrected and changed, we inserted a photo for the production of green walnuts, which already used in feeding honey bees, thanks!
Table 2
HFM is formed in sugar containing food when heated of cooked. Why it is so high the level in 2 and 3 honey? Are these levels dangerous for consumption? The Codex Alimentarius Standard commission has set the maximum limit for HMF in honey at 40 mg/kg (with a higher limit of 80 mg/kg for honeys originating from tropical regions) to ensure that the product has not undergone extensive heating during processing and is safe for consumption. The abbreviation of HMF content is only HydroxyMethylFurfural, remove “The Role of….”
Response: yes, you are right, thanks!
The HMF value of the original honey was good and lower than 5 mg/kg when the samples arrived to the diary factory for the honey fortified yogurt making. To reaching the proper homogeneity and purity the honey samples were heated up for the homogenization, mixing and filtration. It caused an increase in the HMF value. The samples were taken before the making of yogurt-honey mixture.
We found also, the HFM value may be reach to 68.99 or 383.39 or 206.06 mg kg-1 in studies in Malaysia 30.36–56.10 in other studies carried out in Nepal as published by Shapla et al. (2018). 5- Ummay Mahfuza Hydroxymethylfurfural (HMF) levels in honey and other food products: effects on bees and human health. Chemistry Central Journal (2018) 12:35 https://doi.org/10.1186/s13065-018-0408-3

Reviewer 2 Report
Dear colleagues,
It is an interesting study of the effect of including plant extracts in the diet of bees and the use of honey produced by them in a yogurt preparation to lower blood sugar in humans.
The strong point is represented by the innovative idea of ​​obtaining preparations with the potential to improve human health.
My first specific recommendation is that the text to be corrected by a native English speaker (L58 '' of important '' correct is '' is important '', L66 '' which promotes human health as a functional food '' unclear wording, L80 '' healthy '' should be changed to '' health '', L81 '' no reports '' should be '' there are no reports ''). These are just a few examples, the text needs to be corrected in all chapters, including results and discussion.
The paragraph at the end of the introduction stating the objective of the study needs to be improved L83-89 because the wording is unclear.
Fig 1 L 111 in the description please change "The blood sugar level of the participants ..." with "The blood sugar level of each individual participant ...".
Fig 2 L 128 '' Morning blood sugar of the participants before (yellow column) and after (blue column) the human clinical trial. '' Please change with '' Mean values ​​of morning blood sugar of the participants before (blue column) ) and after (yellow column) the human clinical trial. ''
L120-126 the presentation of the results regarding figure 3 is unclear, please reformulate.
Regarding figure 3, if I understood correctly, the values ​​for each individual should correspond to those in figure 1 group 4. In figure 1 we see that individual number 3 has the same values ​​before and after treatment and in figure 3 individual number 1 has the same values. Please re-check and correct if necessary.
Author Response
The authors would like to thanks for Reviewer comments that improve our manuscript quality.
Comments and Suggestions for Authors
Dear colleagues,
It is an interesting study of the effect of including plant extracts in the diet of bees and the use of honey produced by them in a yogurt preparation to lower blood sugar in humans.
The strong point is represented by the innovative idea of obtaining preparations with the potential to improve human health.
Response: Many thanks for your encouraging words, thanks!
My first specific recommendation is that the text to be corrected by a native English speaker (L58 '' of important '' correct is '' is important '',
Response: Many thanks done, thanks!
L66 '' which promotes human health as a functional food '' unclear wording,
Response: Corrected, thanks!
L80 '' healthy '' should be changed to '' health '',
Response: Many thanks done, thanks!
L81 '' no reports '' should be '' there are no reports '').
Response: Many thanks done, thanks!
These are just a few examples, the text needs to be corrected in all chapters, including results and discussion.
Response: ok, thanks!
The paragraph at the end of the introduction stating the objective of the study needs to be improved L83-89 because the wording is unclear.
Response: ok, improved, thanks!
Fig 1 L 111 in the description please change "The blood sugar level of the participants ..." with "The blood sugar level of each individual participant ...".
Response: ok, done, thanks!
Fig 2 L 128 '' Morning blood sugar of the participants before (yellow column) and after (blue column) the human clinical trial. '' Please change with '' Mean values of morning blood sugar of the participants before (blue column) and after (yellow column) the human clinical trial. ''
Response: ok, done, thanks!
L120-126 the presentation of the results regarding figure 3 is unclear, please reformulate.
Response: thanks! Improved and corrected by adding
The data were presented in ascending order in Figure 3, and calculated by subtraction of the morning blood sugar value measured on the 21st day of the trial from the 0-day of the trial.
Regarding figure 3, if I understood correctly, the values for each individual should correspond to those in figure 1 group 4. In figure 1 we see that individual number 3 has the same values before and after treatment and in figure 3 individual number 1 has the same values. Please re-check and correct if necessary.
Response: ok, corrected thanks!
This is the explanation:
The data were presented in ascending order in Figure 3, and calculated by subtraction of the morning blood sugar value measured on the 21st day of the trial from the 0-day of the trial.

Reviewer 3 Report
I appreciate author's attempt to study the impact of consuming yogurt fortified with honey produced by feeding bees on natural plant extract on controlling human blood sugar level. This study is of great importance as it deal with the practical application of functional food product. However, I have certain concerns regarding this manuscript that need to be resolve before publication.
- The language of manuscript and presentation needs an extensive improvement to maintain the interest of readers.
- Line 32, it was mentioned that 150 mL yogurt mixed with 20 ml honey, whereas in Fig 7, it was mentioned that 30 g honey. Be consistent with one units.
- Line 37, here 15 people are mentioned, whereas in Fig 7, different number of participants are shown.
- Line 53, citation style need to be consistent.
- Line 61, provide reference to this statement.
- Line 74-76, rewrite sentence for better understanding.
- Line 78-81, clearly, mentioned about the literature related to fortification of food products with honey especially yogurt and other diary products based on which research gaps were formulated.
- Line 83, Rewrite objectives of present study for better understanding.
- Line 95-97, how these parameters were measured, mention in materials and methods section.
- Discussion section need to be more focus on the findings of present study.
- Line 220-221, mention location as place name, city name, country name.
- Line 241, what is color culture, explain?
- Line 250, mention the city and country where these hospitals are located.
- How the parameters mentioned in Table 2 were determined, mention clearly in materials and methods section.
- In Table 2, why standard deviations of first 7 parameters are not mentioned.
- Line 307, which version of SPSS was used.
Author Response
The authors would like to thanks for Reviewer comments that improve our manuscript quality.
Comments and Suggestions for Authors
I appreciate author's attempt to study the impact of consuming yogurt fortified with honey produced by feeding bees on natural plant extract on controlling human blood sugar level. This study is of great importance as it deal with the practical application of functional food product. However, I have certain concerns regarding this manuscript that need to be resolve before publication.
Response: Many thanks for your so kind words and we are ready for improving the MS to be ready for publication!
- The language of manuscript and presentation needs an extensive improvement to maintain the interest of readers.
Response: OK, the MS has been edited, thanks!
- Line 32, it was mentioned that 150 mL yogurt mixed with 20 ml honey, whereas in Fig 7, it was mentioned that 30 g honey. Be consistent with one unit.
Response: OK, corrected, thanks!
- Line 37, here 15 people are mentioned, whereas in Fig 7, different number of participants are shown.
Response: the starting number of candidates was 15 but during and after carrying out the experiment Some volunteers were excluded from the study as follows: for the 1st, 2nd and 3rd group 1, 3, and 1 volunteer, respectively, because one had a high initial blood sugar level, and the others gave up participation in the trial, whereas in the 4th group, all of the participants took part from the beginning till the end.
- Line 53, citation style needs to be consistent.
Response: we changed according to the system in journal to
[2] IDF (2022) Diabetes Atlas, https://diabetesatlas.org accessed on 28.03.2022
- Line 61, provide reference to this statement.
Response: The ref. was added, thanks!
- Line 74-76, rewrite sentence for better understanding.
Response: rewritten, thanks!
- Line 78-81, clearly, mentioned about the literature related to fortification of food products with honey especially yogurt and other diary products based on which research gaps were formulated.
Response: Thanks, a paragraph was added to the revised MS from line 70 to 73 and 77 to 89.
- Line 83, Rewrite objectives of present study for better understanding.
Response: rewritten, thanks!
- Line 95-97, how these parameters were measured, mention in materials and methods section.
Response: thanks, here the missing methods in details,
this table was added to supplementary (Table S2)
|
WHO code |
Test Name |
INSTRUMENT |
METHOD |
|
21020 |
Determination of total protein in serum |
Siemens Advia XPT |
Biuret method without samples |
|
21040 |
Determination of albumin in serum by the dye - binding method |
Siemens Advia XPT |
Bromocresol green (BCG) method |
|
21072 |
Quantitative determination of reactive protein C (CRP) |
Siemens Advia XPT |
Immunoturbidimetry |
|
21120 |
Determination of urea in serum |
Siemens Advia XPT |
Urease-GLDH method, enzymatic UV test |
|
21130 |
Determination of uric acid |
Siemens Advia XPT |
Uricase / POD / 4-aminoantipyrine ADPS method with ascorbate oxidase (ASOD) |
|
21143 |
Determination of creatinine by enzymatic method |
Siemens Advia XPT |
Enzymatic photometric test (PAP method) |
|
21312 |
Determination of glucose by the hexokinase method |
Siemens Advia XPT |
Hexokinase method |
|
21411 |
Determination of triglycerides |
Siemens Advia XPT |
GPO-PAP method |
|
2142A |
Determination of HDL cholesterol by direct method |
Siemens Advia XPT |
Direct HDL cholesterol method |
|
21420 |
Determination of total cholesterol |
Siemens Advia XPT |
Enzymatic CHOD-PAP method |
|
21422 |
Determination of LDL cholesterol by direct method |
Siemens Advia XPT |
Direct LDL cholesterol method |
|
21500 |
Determination of sodium in serum |
Siemens Advia XPT |
Ion selective electrode (ISE), indirect potentiometry |
|
21501 |
Determination of potassium in serum |
Siemens Advia XPT |
Ion selective electrode (ISE), indirect potentiometry |
|
21510 |
Determination of total calcium |
Siemens Advia XPT |
Ortho-cresolphthalein complexone method |
|
21571 |
Determination of magnesium |
Siemens Advia XPT |
Xylidyl blue color former |
|
22885 |
Determination of alpha-amylase in body fluids |
Siemens Advia XPT |
EPS-G7 (liquid) |
|
24500 |
Determination of lactic acid dehydrogenase (LDH) |
Siemens Advia XPT |
DGKCh method |
|
24600 |
Determination of aspartate aminotransferase (ASAT, GOT) |
Siemens Advia XPT |
IFCC standardized method without pyridoxal phosphate |
|
24610 |
Determination of alanine aminotransferase (ALAT, SGPT) |
Siemens Advia XPT |
IFCC standardized method without pyridoxal phosphate |
|
24620 |
Determination of creatine kinase (CK) |
Siemens Advia XPT |
IFCC reference method |
|
24640 |
Determination of gamma-glutamyltransferase |
Siemens Advia XPT |
IFCC reference method |
|
24700 |
Determination of alpha-amylase in serum |
Siemens Advia XPT |
EPS-G7 (liquid) |
|
24710 |
Determination of lipase |
Siemens Advia XPT |
Enzymatic photometric test (liquid) |
|
24720 |
Determination of alkaline phosphatase |
Siemens Advia XPT |
Standard (DGKCh) method |
|
24741 |
Determination of pseudo-cholinesterase |
Siemens Advia XPT |
Substrate: butyrylthiocholine iodide, DGKCh'94 |
|
2678A |
Determination of IgM |
Siemens Advia XPT |
Turbidimetry |
|
26780 |
Determination of IgG |
Siemens Advia XPT |
Turbidimetry |
|
26788 |
Determination of IgA |
Siemens Advia XPT |
Turbidimetry |
|
28014 |
Blood count with automatic machine IV. |
Sysmex XN 1000 |
Machine counting of electrical / optical pulses |
|
28330 |
Determination of iron-binding capacity |
Siemens Advia XPT |
ADVIA CHEMISTRY TIBC |
|
28350 |
Definition of iron |
Siemens Advia XPT |
Ferrosin / Ferrene method |
|
28360 |
Determination of total transferrin |
Siemens Advia XPT |
Turbidimetry |
|
28390 |
Determination of ferritin |
Siemens Advia XPT |
Immunoturbidimetry |
|
28494 |
Determination of hemoglobin A1c by HPLC, mass spectrometry |
Arkray ADAMS HA-8180 |
Ion exchange chromatography |
|
28610 |
Determination of thrombin time |
Sysmex CS-5100 |
Turbidimetry |
|
28620 |
Determination of prothrombin |
Sysmex CS-5100 |
Turbidimetry |
|
28621 |
Activated partial thromboplastin time |
Sysmex CS-5100 |
Turbidimetry |
- Discussion section need to be more focus on the findings of present study.
Response: thanks, done by added two paragraphs in the revised MS from line 179 to 184 and from 222 to 229.
- Line 220-221, mention location as place name, city name, country name.
Response: Added, thanks!
- Line 241, what is color culture, explain?
Response: changed to starter culture, thanks!
- Line 250, mention the city and country where these hospitals are located.
Response: Added, thanks!
- How the parameters mentioned in Table 2 were determined, mention clearly in materials and methods section.
Response: the methodology below, thanks
The basic parameters: water content, Fructose +glucose, Fructose, Glucose, Free acid content, HMF content, Diastase activity were determined according to the HARMONISED METHODS OF THE INTERNATIONAL HONEY COMMISSION (IHC 2009) http://www.bee-hexagon.net/en/network.htm.
The concentrations of boron, calcium, copper, iron, potassium, magnesium, sodium, phosphorus, sulphur, and zinc were determined by ICP-OES (Inductively Coupled Plasma Optical Emission Spectrometer) (Thermo Scientific iCAP 6300, Cambridge, UK) according to Czipa et al. (2019).
- In Table 2, why standard deviations of first 7 parameters are not mentioned.
Response: added and corrected, thanks!
- Line 307, which version of SPSS was used.
Response: thanks, SPSS Statistics V22.0 added to the revised MS.

Round 2
Reviewer 2 Report
Dear colleagues, the second version is revised accordingly, so I have only a few suggestions: to detail the results regarding the chemical analysis of honey, this being the bioactive compound that is the basis of this study. This aspect should also be detailed in the discussion chapter. In paragraph L162-164 to include a citation to support the statement. Congratulations on your study!
Author Response
The authors would like to thanks for Reviewer comments that improve our manuscript quality.
Comments and Suggestions for Authors
Dear colleagues, the second version is revised accordingly, so I have only a few suggestions: to detail the results regarding the chemical analysis of honey, this being the bioactive compound that is the basis of this study. This aspect should also be detailed in the discussion chapter. In paragraph L162-164 to include a citation to support the statement. Congratulations on your study!
Response: Many thanks for your great words!
The part was added to revised MS, thanks!

Reviewer 3 Report
The quality of the manuscript is improved significantly after revision. I recommend publication of manuscript after slight improvements.
- Title of the manuscript can be rewritten as "Functional Yogurt Fortified with Honey produced by Feeding Bees on Natural Plant Extracts for Controlling Human Blood Sugar Level".
- The statistical significant differences among the mean values of various parameters in Table 2 need to be presented.
Author Response
The authors would like to thanks for Reviewer comments that improve our manuscript quality.
Comments and Suggestions for Authors
The quality of the manuscript is improved significantly after revision. I recommend publication of manuscript after slight improvements.
- Title of the manuscript can be rewritten as "Functional Yogurt Fortified with Honey produced by Feeding Bees on Natural Plant Extracts for Controlling Human Blood Sugar Level".
- The statistical significant differences among the mean values of various parameters in Table 2 need to be presented.
Response: Many thanks! The title is already changed as you suggested, thanks again!
The statistical analysis in Table 2 was added to the revised MS, thanks!
